# Social Support Mediates the Relationship between COVID-19-Related Burnout and Booster Vaccination Willingness among Fully Vaccinated Nurses

**DOI:** 10.3390/vaccines11010046

**Published:** 2022-12-25

**Authors:** Petros Galanis, Aglaia Katsiroumpa, Panayota Sourtzi, Olga Siskou, Olympia Konstantakopoulou, Theodoros Katsoulas, Daphne Kaitelidou

**Affiliations:** 1Clinical Epidemiology Laboratory, Faculty of Nursing, National and Kapodistrian University of Athens, 11527 Athens, Greece; 2Faculty of Nursing, National and Kapodistrian University of Athens, 11527 Athens, Greece; 3Department of Tourism Studies, University of Piraeus, 18534 Piraeus, Greece; 4Center for Health Services Management and Evaluation, Faculty of Nursing, National and Kapodistrian University of Athens, 11527 Athens, Greece

**Keywords:** COVID-19, burnout, social support, willingness, nurses, vaccination

## Abstract

COVID-19 booster doses for high-risk groups such as nurses are necessary to reduce the impacts of the pandemic and promote public health. We examined the relationship between COVID-19-related burnout and booster vaccination willingness among nurses, and we assessed whether social support can buffer this relationship. We conducted a cross-sectional study with 963 fully vaccinated nurses working in healthcare settings in Greece. We used the multidimensional scale of perceived social support to measure social support and the COVID-19 burnout scale to measure COVID-19-related burnout. We measured vaccination willingness with a scale from 0 (extremely unlikely to take a booster dose) to 10 (extremely likely). Among nurses, 37.1% reported being very likely to be vaccinated, 34.4% reported being uncertain about their likelihood of vaccination, and 28.6% reported being very unlikely to be vaccinated with a booster dose. We found that COVID-19-related burnout reduced vaccination willingness, while social support functioned as a partial mediator of this relationship. In conclusion, nurses who experienced burnout were less likely to accept a booster dose. Furthermore, increasing nurses’ social support reduced the negative effects of burnout, resulting in improved booster vaccination willingness. Immunization awareness programs should be implemented in order to address nurses’ concerns and support booster doses.

## 1. Introduction

SARS-CoV-2 continues to threaten global health 3 years after its first report in China [1]. COVID-19 has caused more than 646 million confirmed cases and 6.64 million deaths worldwide as of 27 November 2022 [2]. Moreover, a new syndrome has emerged among COVID-19 survivors, i.e., post-COVID-19 syndrome or long-term COVID-19, causing a wide variety of clinical manifestations such as fatigue, cough, dyspnea, sleep disorders, anxiety, depression, myalgia, deep-vein thrombosis, pulmonary embolism, and acute myocardial infarction [3,4,5].

Although 68.5% of the world population has received at least one dose of a vaccine [6], the emergence of several SARS-CoV-2 variants and the fact that COVID-19 vaccines or previous infection produce incomplete immunity against new variants (e.g., the Delta and Omicron variants) [7,8] necessitate further booster doses. A booster dose reduces SARS-CoV-2 infection rates but effectiveness against the Omicron variant is significantly less than that against the Delta variant [9,10]. As SARS-CoV-2 is anticipated to change regularly in the future, timely immunization with yearly booster doses, especially for vulnerable groups, seems to be the best strategy to strengthen our immunity against the virus [11,12,13].

Healthcare workers are at greater risk of infection from SARS-CoV-2 than the general population. In particular, healthcare workers are more than 10 times more likely to be infected with the virus than the general public [14]. Moreover, the literature suggests increased COVID-19-associated morbidity and mortality among healthcare workers [15,16]. Therefore, several committees worldwide have already suggested the approval for further booster doses for healthcare workers [17,18]. However, as of 27 November 2022, among 78.6 million healthcare workers that completed the primary vaccination series, only 32.4 million received a booster dose [19].

The COVID-19 vaccination program has been a success so far, but future success could be undermined by a decreased willingness of individuals to accept booster doses. Lower booster uptake among nurses is the worst scenario since their negative attitude toward booster vaccination could affect decisions of the general public and lead to negative consequences in society [20]. Until now, only one study has investigated the intention of nurses to take a booster dose, finding that 69.1% of fully vaccinated nurses are willing to accept a booster dose [21]. A recent meta-analysis found that the acceptance of a booster dose among healthcare workers was 66%, with higher levels of acceptance in the Western Pacific region and the European region and lower levels in the Southeast Asian region and the Eastern Mediterranean region [22]. Moreover, the acceptance rate among healthcare workers (66%) was lower than that of the general public (81%).

Several systematic reviews and meta-analyses during the COVID-19 pandemic confirmed the fact that nurses have experienced high levels of burnout [23,24,25]. Sociodemographic (gender, age, and clinical experience), work-related (workload, longer working time, and inadequate and insufficient material and human resources), psychological (social support and coping strategies), and COVID-19-related (perceived threat of COVID-19) factors affect burnout among nurses. Moreover, the literature suggests that levels of burnout among nurses are higher during the pandemic than those before the pandemic [26]. However, all these studies considered burnout as an occupational phenomenon that results from chronic workplace stress without successful management [27]. To the best of our knowledge, there are no studies until now that measured nurses’ burnout in the context of pandemic, i.e., COVID-19-related burnout. Since nurses’ burnout is associated with negative outcomes such as depression, poor quality of life, sleep disorders, poor quality of care, and reduced patient satisfaction, organizational commitment, and productivity [28,29,30,31], we hypothesized the following:

**H1:** 
*COVID-19-related burnout among nurses significantly and negatively predicts booster vaccination willingness (Figure 1).*


Social support is defined as “support accessible to an individual through social ties to other individuals, groups, and the larger community” [32]. Social support during the COVID-19 pandemic is essential because people have experienced negative psychological effects including social isolation, loneliness, post-traumatic stress disorder, anger, and confusion due to repeated lockdowns and quarantine measures [33,34]. Several studies support the positive impact of social support on people’s mental health (e.g., anxiety, stress, depression, and post-traumatic stress disorder) during the pandemic either directly [35,36,37,38] or indirectly [39,40,41]. Moreover, Jaspal and Breakwell (2022) discovered that social support was a determinant of COVID-19 vaccination intention and testing [42]. Furthermore, Stickley et al. (2021) found that loneliness (an indicator of decreased social support) was inversely associated with engaging in COVID-19 preventive attitudes, such as hand-washing, wearing a face mask, and social distancing [43]. The literature supports the positive effect of social support on nurses’ life during the pandemic either directly as a determinant [44] or indirectly as a mediator [45,46,47]. Therefore, on the basis of the current literature, we hypothesized the following:

**H2:** 
*Social support functions as a significant mediator in the association between COVID-19-related burnout and vaccination willingness (Figure 1).*


Lastly, several studies [48,49,50,51,52,53] investigated the impact of demographic data (e.g., gender, age, education level, and chronic disease) and COVID-19-related variables (e.g., previous COVID-19 infection, perceived risk, and fear about COVID-19) on nurses’ willingness to accept primary COVID-19 vaccination, but only one study [21] assessed this willingness toward booster doses. Therefore, we hypothesized the following:

**H3:** 
*Demographic data (gender, age, education level, chronic condition, clinical experience, and self-assessment of health status) and COVID-19-related variables (COVID-19 infection, booster dose, and side-effects because of COVID-19 vaccination) could have an impact on nurses’ vaccination willingness (Figure 1).*


## 2. Materials and Methods

### 2.1. Study Design

Our cross-sectional study was conducted in Greece between 18 September 2022 and 18 October 2022. All nurses working in healthcare settings in Greece and understanding the Greek language were eligible to participate in our study. Nurses that had not been vaccinated with the primary dose could not participate in our study since the Greek Health Ministry offered from 14 September 2022 a new booster dose only to fully vaccinated healthcare workers.

We created an online Greek language survey tool using Google forms. We distributed the survey tool through a nurse group on social media. Moreover, we reached out to our networks of contacts in order to recruit nurses. We asked agreeable nurses to recommend other potential nurses from among their networks, applying the snowball technique. Finally, we achieved a convenience sample.

We did not collect personally identifying information, and responses were anonymized. All participants gave their informed consent prior to enrollment in the study. In particular, we asked participants to accept an informed consent form and then we allowed them to access the full survey tool. Our study was conducted according to the guidelines of the Declaration of Helsinki and approved by the Ethics Committee of Faculty of Nursing, National and Kapodistrian University of Athens (reference number; 370, 02-09-2021).

We used the rule of thumb given by Hair et al. (2017) in order to estimate the sample size for mediation analysis in our study [54]. Therefore, the required sample size was 250 nurses (= 25 variables × 10 = 250) since the number of participants should be at least 10 times that of study variables. Moreover, considering that the target population of nurses in Greece is about 27,103 employees, we needed at least 651 nurses with a confidence level of 99%, a margin of error of 5%, and a prevalence of 50% [55,56]. We chose to increase our sample in order to achieve a higher precision level of our measurements.

### 2.2. Measures

#### 2.2.1. Demographic and COVID-19-Related Variables

We collected demographic data of nurses, including gender, age, MSc/PhD diploma, years of experience, chronic condition, and self-assessment of health status (very poor, poor, moderate, good, and very good). Furthermore, we collected COVID-19-related data such as COVID-19 infection (no or yes), booster doses, and side-effects because of COVID-19 vaccination (a scale from 0 [none] to 10 [many]).

#### 2.2.2. Booster Vaccination Willingness

We used the following question in order to measure the willingness of nurses to accept a booster dose: “A new booster dose against COVID-19 is suggested for all fully vaccinated nurses with primary doses. This booster dose is not compulsory. How likely is it that you will take a booster dose?”. We measured vaccination willingness on an 11-point scale (0 = extremely unlikely and 10 = extremely likely). In order to categorize nurses according to their score on vaccination willingness scale, we used a priori cutoff points. In particular, we considered nurses with scores ≤2 as “very unlikely to be vaccinated”, nurses with scores of 3–7 as “uncertain to be vaccinated”, and nurses with scores ≥8 as “very likely to be vaccinated”.

#### 2.2.3. Social Support

We used the multidimensional scale of perceived social support (MSPSS) to measure subjectively assessed social support [57]. MSPSS consists of 12 items (e.g., “There is a special person who is around when I am in need”), and answers are in a seven-point Likert scale from 1 (strongly disagree) to 7 (strongly agree). Total score for the MSPSS ranges from 1 (low social support) to 7 (high social support). MSPSS includes support from family, friends, and significant others. MSPSS has been validated in Greek [58]. In our study, Cronbach’s alpha for MSPSS was 0.94.

#### 2.2.4. Burnout

The COVID-19 burnout scale (COVID-19-BS) was used to measure COVID-19-related burnout [59]. The scale consists of 13 items (e.g., “I feel tired applying personal protection measures, e.g., wearing a face mask”). Answers are measured on a five-point scale (1 = strongly disagree, 5 = strongly agree). The overall COVID-19-BS score ranges from 1 (low level of burnout) to 5 (high level of burnout). The scale manifested excellent reliability in our study (Cronbach’s alpha = 0.912).

### 2.3. Statistical Analysis

We used descriptive statistics to present our variables: numbers and percentages for categorical variables, and mean and standard deviation for continuous variables. Scores on COVID-19-BS, MSPSS, and vaccination willingness scale followed a normal distribution. Correlations between continuous variables and COVID-19-BS, MSPSS, and vaccination willingness scale were estimated using Pearson’s correlation coefficient. Prior to the mediation analysis, we constructed a multivariable linear regression model in order to assess the impact of demographic data and COVID-19-related variables on nurses’ willingness. In that case, we presented the adjusted coefficient beta, 95% confidence intervals (95% CI), and *p*-values. Additionally, in order to check independence of observations, homoscedasticity, and collinearity in the multivariable model, we used standardized residual plots, the tolerance test, the variance inflation factor, and the Durbin–Watson statistic. Acceptance values were >0.5 for the tolerance test, <4.0 for the variance inflation factor, and about 2.0 for the Durbin-Watson statistic [60]. Since age and clinical experience violated the assumptions of linear regression analysis, we decided to remove clinical experience from the model and keep age since the literature suggests a possible impact of age on nurses’ willingness. The PROCESS macro (Model 4) was used to test the mediating effect of social support in the relationship between COVID-19-related burnout and vaccination willingness [61]. We used the 95% CI to examine the significance of effects based on 5000 bootstrap samples [62]. If the 95% CI did not include zero, then the effect was significant. Moreover, we calculated regression coefficients (β) and squared multiple correlations (R^2^). All statistical tests were conducted at α = 0.05. IBM SPSS 21.0 (IBM Corp. Released 2012. IBM SPSS Statistics for Windows, Version 21.0. Armonk, NY: IBM Corp.) was used for statistical analysis.

## 3. Results

### 3.1. Sample Characteristics

The final sample consisted of 963 nurses aged 21–63 years (mean = 37.9, standard deviation = 9.6). Most of the nurses were females (88.4%). More than half of the nurses possessed a MSc/PhD diploma (54.6%), while their mean clinical experience was 12 years (standard deviation = 9.2). Most nurses were infected with SARS-CoV-2 during the pandemic (71.8%) and received a booster dose (89.4%). Among the nurses, 87.3% experienced side-effects because of a past COVID-19 vaccine dose. Table 1 shows the characteristics of the nurses in our study.

### 3.2. Descriptive Statistics and Correlations

Descriptive statistics of continuous variables and COVID-19-BS, vaccination willingness score, and MSPSS are shown in Table 2. The mean vaccination willingness score was 5.3 indicating a moderate level of nurses’ willingness to take a booster dose. Moreover, 37.1% of nurses (n = 357) reported being very likely to be vaccinated, 34.4% (n = 331) reported being uncertain about their likelihood of vaccination, and 28.6% (n = 275) reported being very unlikely to be vaccinated with a booster dose. Mean score on MSPSS indicated a high level of support, while mean score on COVID-19-BS indicate a moderate level of burnout.

Correlation analysis between continuous variables is shown in Table 3. Pearson correlation analysis showed that COVID-19-related burnout was negatively related to vaccination willingness (r = −0.291, *p* < 0.001) and total social support (r = −0.183, *p* < 0.001). In addition, total social support was positively related to vaccination willingness (r = 0.136, *p* < 0.001).

### 3.3. Linear Regression Analysis

Multivariable linear regression analysis, with vaccination willingness score as the dependent variable and demographic data and COVID-19-related variables as the independent variables, is shown in Table 4. Male sex, older age, nurses with a chronic condition, non-COVID-19 infection, booster dose, fewer side-effects because of COVID-19 vaccination, less COVID-19-related burnout, and more social support were independent predictors of nurses’ intention to accept booster vaccination.

### 3.4. Mediation Analysis

Hypothesis 1 assumes that COVID-19-related burnout decreases booster vaccination willingness. We confirmed this hypothesis since we found that COVID-19-related burnout significantly predicted vaccination willingness (β = −0.8922, *p* < 0.0001). COVID-19-related burnout explained 8.4% of the variance in vaccination willingness. Moreover, our results showed that COVID-19-related burnout decreases social support (β = −0.2205, *p* < 0.0001).

Hypothesis 2 assumes that social support mediates the relationship between COVID-19-related burnout and vaccination willingness. We confirmed hypothesis 2 since COVID-19-related burnout had a significant indirect effect on vaccination willingness through social support (β = 0.0437, 95% CI = 0.0041 to 0.0922, standard error = 0.0223). The bootstrap 95% CI for the mediating effect of social support on vaccination willingness did not include zero; thus, there was a significant association between these two variables. COVID-19-related burnout and social support accounted for 23.0% of the total effect.

Therefore, the two hypotheses were supported by our data. Mediation analysis showed that social support partly mediates the relationship between COVID-19-related burnout and vaccination willingness (Figure 2). Detailed results from mediation analysis are summarized in Table 5.

## 4. Discussion

To the best of our knowledge, this is the first study to evaluate the direct and indirect impact of COVID-19-related burnout among nurses on booster vaccination willingness through social support. Our results confirm the three study hypotheses. In particular, we found that burnout reduces vaccination willingness, but social support buffers this negative relationship. Moreover, we found that several demographic data and COVID-19-related variables affect nurses’ willingness.

Among nurses in our study, only 37.1% reported that they intend to accept a future booster dose, while 34.4% were hesitant and 28.6% had a negative attitude toward a new booster dose. Since the majority of nurses in our sample had already taken a booster dose (89.4%), the percentage of nurses that reported willingness to accept a new booster dose (37.1%) is worrying. A similar study in Greece was conducted in May 2022 and found that 69.1% of nurses intended to take a second booster dose [21]. Furthermore, according to a meta-analysis, the COVID-19 first booster dose acceptance among healthcare workers was 66% with a wide range from 36% to 90% [22]. However, all studies in this meta-analysis measured the willingness of healthcare workers to accept a first booster dose and were conducted much earlier than our study, i.e., from August 2021 to February 2022. In addition, vaccination willingness was measured among all healthcare workers in clinical settings; thus, authors did not calculate willingness rate separately for physicians, nurses, paramedical staff, etc. Therefore, although direct comparisons between previous studies and our study are unsafe, it seems to be that, as time passes, healthcare workers’ willingness to take further booster doses reduces.

We found that COVID-19-related burnout is negatively associated with booster vaccination willingness. Since there are no studies regarding the relationship between COVID-19-related burnout and vaccination willingness among nurses, it is impossible to discuss our finding with the literature. However, an increased number of COVID-19 vaccine doses, side-effects from previous COVID-19 doses, and misinformation could result in the “vaccine fatigue” phenomenon [63]. In the case of nurses, the situation is even more complicated since they have the moral obligation to be vaccinated in order to create a safe healthcare environment and protect their patients. Moreover, most of the countries have placed healthcare workers high on the priority list for COVID-19 vaccines in order to keep the healthcare system running. Thus, nurses have been working under tremendous pressure during the pandemic, experiencing high level of burnout [23]. In addition, job burnout among nurses causes several other issues during the pandemic such as poor quality of life, depression, anxiety, sleep disorders, reduced productivity, and poor quality of care [28,29,30,31]. Levels of mental and physical fatigue among nurses are high 3 years after the onset of the pandemic, and this exhaustion can negatively affect their attitudes toward preventive measures, e.g. wearing masks, hand washing, and booster vaccination.

Additionally, we found a negative relationship between COVID-19-related burnout and social support. This finding is supported by several previous studies where burnout was negatively correlated with occupational burnout [64,65,66]. Furthermore, a recent systematic review found that supervisor and coworker support reduces nurses’ burnout [67]. Social support can help nurses to develop their positive capacity and promote their internal positive competence and self-efficacy [68]. Social support meets the basic human demands of affiliation and nurses seek help from family members, friends, and significant others in order to confront psychological difficulties [69].

As hypothesized, our findings showed that social support buffered the negative relationship between COVID-19-related burnout and vaccination willingness. Thus, social support could be a protective factor helping nurses to cope more effectively with the negative effects of burnout. Nurses’ encouragement or approval could result in a decrease in burnout. Therefore, increasing nurses’ social support reduces the negative effects of COVID-19-related burnout, resulting in improved vaccination willingness. Evidence supports our findings since several studies during the pandemic confirmed the positive impact of social support on nurses’ life as a mediator [45,46,47]. In particular, Du et al. (2022) found that perceived organizational support is a mediating variable between occupational stress and insomnia symptoms [45], while Fronda et al. (2022) found that adequate social support as a mediating factor could partially reduce the effects of coronaphobia on professional turnover intention in nurses [46]. In addition, perceived social support improved the protective role of resilience against the anxiety that nurses experience during the pandemic [47]. Moreover, recent studies confirmed the positive impact of social support in countering psychological issues since support by coworkers and managers reduced the negative mental health consequences caused by the pandemic [70,71]. Therefore, social support is essential to mitigate or alleviate the negative effects of variables such as occupation stress, emotional stress, and coronaphobia and improve mental health among nurses.

We found that male sex, older age, and chronic condition were independent demographic predictors of booster vaccine intention among nurses in our study. The literature confirms these findings since several studies found that males, older nurses, and those with a chronic condition had a higher intention to accept primary COVID-19 vaccination and booster doses [21,48,49,50,72]. It is well known that COVID-19-related severe outcomes such as hospitalization, mechanical ventilation, and mortality are higher among males, the elderly, and individuals with comorbidity [73,74,75]. Thus, it is probable that perceived risk of COVID-19 and fear about the disease are higher among nurses that belong to high-risk groups, leading to a higher acceptance rate of booster doses.

In addition, we found that nurses who experienced more side-effects from previous COVID-19 vaccination were less likely to accept further booster doses. Evidence has shown that primary COVID-19 intention is lower among nurses who have concerns about side-effects and the safety of COVID-19 vaccines [49,50,51]. Moreover, increased fear of COVID-19 vaccines and less confidence in vaccination effectiveness reduced nurses’ willingness to accept COVID-19 vaccination [21,48,52]. Therefore, side-effects from previous COVID-19 vaccination might shake nurses’ faith in COVID-19 vaccines, acting as a barrier to the acceptance of future booster doses. Fortunately, side-effects after the COVID-19 vaccination are usually mild and self-limited [76,77,78]. Since lack of knowledge about COVID-19 vaccines reduced nurses’ willingness to accept vaccination [49], policymakers should inform nurses that the benefits of COVID-19 vaccines outweigh the risks.

### Limitations

Our study had some limitations. Firstly, we used a convenience sample; thus, our results may be of limited representativeness. A nationally representative sample would add invaluable evidence on the topic. Secondly, we could not establish causal relationships due to the cross-sectional study design. Moreover, burnout, social support, and vaccination willingness among nurses may change over time. Longitudinal studies could document nurses’ attitudes in a more valid way. Thirdly, we used self-reported questionnaires to measure nurses’ attitudes and a self-reporting bias is probable. Future studies should measure booster uptake among nurses in order to understand their actual decision. Fourthly, we conducted our study in September and October of 2022 when confirmed COVID-19 cases were relatively low in Greece. Therefore, the perceived risks of the pandemic may have influenced the willingness of nurses to accept a booster dose. Fifthly, among many variables, we only evaluated social support as a possible mediator. Therefore, it is crucial to explore the role of other mediating variables in the relationship between COVID-19-related burnout and booster vaccination willingness among nurses. Lastly, we measured the impact of several sociodemographic variables (i.e., gender, age, education level, clinical experience, chronic condition, and health status) and COVID-19-related variables (i.e., COVID-19 infection during the pandemic, booster doses, and side-effects because of COVID-19 vaccination) on nurses’ willingness to accept a booster dose, but future research could investigate more variables such as COVID-19 positivity during a time interval prior to the study, wearing a mask at work and outside of the workplace, and perceived risk of SARS-CoV-2.

## 5. Conclusions

According to our results, nurses who experienced COVID-19-related burnout were less likely to accept a booster dose. Moreover, increasing nurses’ social support reduced the negative effects of COVID-19-related burnout, resulting in improved booster vaccination willingness. Three years after the onset of the pandemic, burnout could threaten vaccination programs, reducing COVID-19 booster uptake even among nurses. The vaccination attitudes of nurses are crucial since they can protect themselves, their families, and their patients. Moreover, nurses should act as COVID-19 vaccine enablers and communicators to the general public since a negative attitude of nurses toward booster doses may undermine public trust in COVID-19 vaccines. There is a need for timely vaccination with booster doses for high-risk groups such as nurses and vulnerable groups such as the elderly in order to significantly reduce the impacts of COVID-19 pandemic. Policymakers should develop and implement immunization awareness programs in order to address nurses’ concerns and support booster doses.

## Figures and Tables

**Figure 1 vaccines-11-00046-f001:**
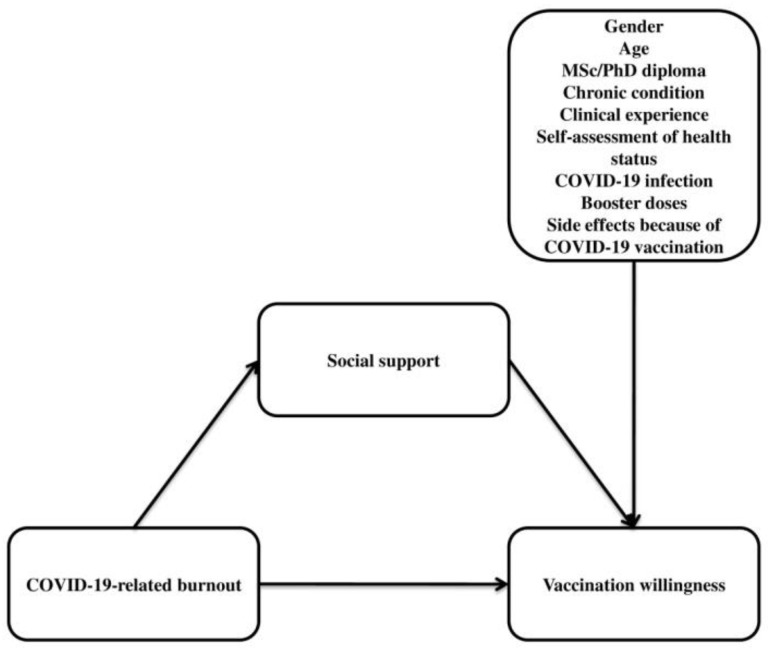
Hypothesized model of the mediation effect of social support on the relationship between COVID-19-related burnout and vaccination willingness, and the impact of demographic data and COVID-19-related variables.

**Figure 2 vaccines-11-00046-f002:**
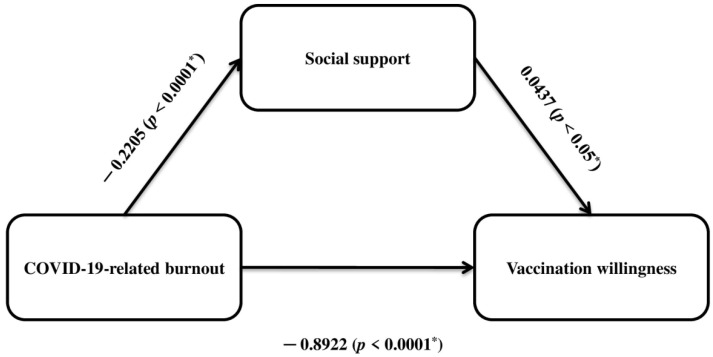
Mediation model of social support on the relationship between COVID-19-related burnout and vaccination willingness. * Statistically significant.

**Table 1 vaccines-11-00046-t001:** Summary of nurses’ characteristics (N = 963).

Variable	N	%
Gender		
Females	851	88.4
Males	112	11.6
MSc/PhD diploma		
No	437	45.4
Yes	526	54.6
Chronic condition		
No	722	75.0
Yes	241	25.0
Self-assessment of health status		
Very poor	26	2.7
Poor	16	1.7
Moderate	70	7.3
Good	580	60.2
Very good	271	28.1
COVID-19 infection		
No	272	28.2
Yes	691	71.8
Booster doses		
No	102	10.6
Yes	861	89.4

**Table 2 vaccines-11-00046-t002:** Descriptive statistics of continuous variables.

Variable	Mean	SD	Median
Age	37.9	9.6	37.0
Clinical experience (years)	12.0	9.2	11.0
Side-effects because of COVID-19 vaccination	3.1	2.6	2.0
COVID-19 vaccination willingness	5.3	3.6	5.0
COVID-19-related burnout	3.2	0.9	3.2
Total social support	6.0	1.2	6.4
Family support	6.0	1.4	6.5
Friends support	5.9	1.4	6.3
Significant others support	6.1	1.3	6.8

SD: standard deviation.

**Table 3 vaccines-11-00046-t003:** Correlation matrix between continuous variables.

Variable	1	2	3	4	5	6	7	8	9
1. Age	-	0.878 ***	0.010	0.152 ***	−0.015	−0.098 **	−0.025	−0.127 ***	−0.108 **
2. Clinical experience (years)		-	0.075 *	0.137 ***	0.006	−0.087 *	−0.028	−0.091 **	−0.112 ***
3. Side-effects because of COVID-19 vaccination			-	−0.240 ***	0.274 ***	−0.047	−0.056	−0.019	−0.049
4. COVID-19 vaccination willingness				-	−0.291 ***	0.136 ***	0.114**	0.141 ***	0.165 ***
5. COVID-19-related burnout					-	−0.183 ***	−0.145 ***	−0.159 ***	−0.175 ***
6. Total social support						-	0.862 ***	0.853 ***	0.892 ***
7. Family support							-	0.554 ***	0.680 ***
8. Friend support								-	0.669 ***
9. Significant other support									-

* *p* < 0.05; ** *p* < 0.01; *** *p* < 0.001.

**Table 4 vaccines-11-00046-t004:** Multivariable linear regression analysis with booster vaccination willingness as the dependent variable.

Independent Variable	Adjusted Coefficient Beta	95% CI for Beta	*p*-Value	Tolerance	VIF
Males vs. females	1.882	1.227 to 2.538	<0.001	0.923	1.083
Age	0.040	0.018 to 0.062	<0.001	0.906	1.104
MSc/PhD diploma	0.274	−0.138 to 0.686	0.193	0.967	1.034
Chronic condition	0.722	0.232 to 1.211	0.004	0.907	1.103
Very poor/poor/moderate health status vs. good/very good	0.614	0.035 to 1.263	0.064	0.941	1.062
Non-COVID-19 infection	0.699	0.233 to 1.164	0.003	0.928	1.077
Booster dose	2.014	1.346 to 2.682	<0.001	0.964	1.037
Side effects because of COVID-19 vaccination	−0.210	−0.293 to −0.126	<0.001	0.879	1.137
COVID-19-related burnout	−0.878	−1.102 to −0.653	<0.001	0.884	1.131
Total social support	0.194	0.040 to 0.348	0.013	0.902	1.109

CI: confidence interval. Adjusted R^2^ for the model = 21.1%; *p*-value for ANOVA < 0.001; Durbin–Watson index = 1.993.

**Table 5 vaccines-11-00046-t005:** Mediation model summary information for the hypothesized mediation model.

	b	SE	t	*p*	95% Confidence Interval	
					LLCI	ULCI
**Direct effects**						
COVID-19-RB → SS	−0.2205	0.0404	−5.46	<0.0001	−0.2998	−0.1412
COVID-19-RB → VW	−0.8922	0.1125	−7.93	<0.0001	−1.1129	−0.6715
**Indirect effect**						
COVID-19-RB → SS → VW	0.0437	0.0223	2.33	<0.05	0.0041	0.0922
**Total effect**	−0.8485	0.1110	-7.65	<0.0001	−1.0663	−0.6307

COVID-19-RB: COVID-19-related burnout; LLCI: lower level of the confidence interval; SS: social support; ULCI: upper level of the confidence interval; VW: vaccination willingness.

## Data Availability

The data presented in this study are available on request from the corresponding author.

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
