# Peer review of "Social Support Mediates the Relationship between COVID-19-Related Burnout and Booster Vaccination Willingness among Fully Vaccinated Nurses"

_vaccines, 2022, doi:10.3390/vaccines11010046_

Round 1

Reviewer 1 Report

I read the manuscript entitled "Social support mediates the relationship between COVID-19-related burnout and booster vaccination willingness among fully vaccinated nurses". The topic of the study is timely and very interesting. However, the statistical analysis seems incomplete and the presentation has significant problems.

I hope the following comments will improve the quality of the manuscript.

1. Introduction

Hypothesis H1 seems redundant as it will be confirmed or not by the verification process of hypothesis H2. Instead, hypotheses are absent on the effect of other parameters such as Gender or Clinical experience, or even COVID-19 infection that may exert Mediation or Moderator and influence the mediation of social support.

2.1. Study design

The study is also epidemiological, so the sample size calculation should be based on the total size of the nursing population. If the recent sample calculations by Sikaras et al. are correct (PMID: 35071671 and PMID: 35052297) the required sample is about 650 people. The sample size of the present study is sufficient, the sample calculation needs improvement.

2.3 Statistical analysis

Normality is one of the assumptions for linear regression, were the others (such as homoscedasticity, independence) were examined?

I would suggest the authors before proceeding to test for mediation to conduct multiple regression to check which other variables explain the variation in Vaccination willingness.

3. Results

Table 1: Please transfer to table 2 the mean values and standard deviations of the continuous variables'

Table 2: Please transfer the correlations to a separate table; it is necessary to add the means and standard deviations of the subscales of social support.

Add a separate table with the correlations of all continuous variables and subscales.

Table 3: Table 3 is really impossible to read, I would suggest replacing it with one like the one attached.

Figure 2:  I would ask you to correct it.

Author Response

Dear Reviewer,

Thank you for giving us the opportunity to revise our manuscript entitled "Social support mediates the relationship between COVID-19-related burnout and booster vaccination willingness among fully vaccinated nurses". We would also like to thank you for your insightful comments and suggestions on how to improve our manuscript. We respectfully tried to address the issues raised and to revise our manuscript accordingly. We hope that our revision will reach the high standards of the journal “Vaccines”. Please, see the attached file. Also, we made changes in the manuscript according to the other Reviewers’ instructions.

We look forward to hearing from you

Best Regards

The authors

Reviewer 2 Report

I am concerned that the authors did not consider whether the individuals had COVID at the time of the questionnaire. Did they ask if they had had COVID within a certain period of time? The tables shows COVID-19 infection I am assuming that they are asking "have you ever had COVID-19". 

Although the concept of burnout and nurses' attitude regarding vaccination is worthwhile the findings are not conclusive without adjusting or looking at COVID-19 positivity during a time interval prior to the questionnaire. I would also expect questions related to wearing a KN95 at work and outside of the workplace as well as perceived risk of exposure. 

Author Response

(The authors gave the same response as above.)

Round 2

Reviewer 2 Report

Ideally recent infection would have been an indicator as well. Otherwise , comments were comments were adequately addressed.